# Synthcity: a benchmark framework for diverse use cases of tabular synthetic data

**Zhaozhi Qian**
University of Cambridge
zq224@cam.ac.uk

**Rob Davis**
University of Cambridge
rd473@cam.ac.uk

**Mihaela van der Schaar**
University of Cambridge
mv472@cam.ac.uk

## Abstract

Accessible high-quality data is the bread and butter of machine learning research, and the demand for data has exploded as larger and more advanced ML models are built across different domains. Yet, real data often contain sensitive information, subject to various biases, and are costly to acquire, which compromise their quality and accessibility. Synthetic data have thus emerged as a complement, sometimes even a replacement, to real data for ML training. However, the landscape of synthetic data research has been fragmented due to the large number of data modalities (e.g., tabular data, time series data, images, etc.) and various use cases (e.g., privacy, fairness, data augmentation, etc.). This poses practical challenges in comparing and selecting synthetic data generators in different problem settings. To this end, we develop Synthcity, an open-source Python library that allows researchers and practitioners to perform one-click benchmarking of synthetic data generators across data modalities and use cases. In addition, Synthcity's plug-in style API makes it easy to incorporate additional data generators into the framework. Beyond benchmarking, it also offers a single access point to a diverse range of cutting-edge data generators. Through examples on tabular data generation and data augmentation, we illustrate the general applicability of Synthcity, and the insight one can obtain.

## 1 Introduction

Access to high quality data is the lifeblood of AI. Although AI holds strong promise in numerous high-stakes domains, the lack of high-quality datasets creates a significant hurdle for the development of AI, leading to missed opportunities. Specifically, three prominent issues contribute to this challenge: *data scarcity*, *privacy*, and *bias* [Mehrabi et al., 2021, Gianfrancesco et al., 2018, Tashea, 2017, Dastin, 2018]. As a result, the dataset may not be available, accessible, or suitable for building performant and socially responsible AI systems [Sambasivan et al., 2021].

This challenge is especially prominent for tabular datasets, which are often curated in highly regulated industries including healthcare, finance, manufacturing etc. Synthetic tabular data has the potential to fuel the development of AI by unleashing the information in datasets that are small, sensitive or biased. To achieve this, we need high-performance generative models that both faithfully capture the data distribution and satisfy additional constraints for the desired use cases.

To date, the landscape of synthetic data research has been fragmented because the combination of *use cases* (i.e. fairness, privacy, and augmentation) and *data modalities* (e.g. static tabular data, time series data, etc.) creates a plethora of problem settings. In response to the large problem space, the

Submitted to the 37th Conference on Neural Information Processing Systems (NeurIPS 2023) Track on Datasets and Benchmarks. Do not distribute.

community has taken a divide-and-conquer approach: highly-specialized generative models have been developed to fit in one particular setting. This has led to a proliferation of specialized generative models [Jordon et al., 2018, Yoon et al., 2020, Ho et al., 2021, Mehrabi et al., 2021, van Breugel et al., 2021, Zhu et al., 2017, Yoon et al., 2018, Saxena and Cao, 2021].

This fragmented landscape has created four main challenges for benchmarking synthetic data generators, which would hamper the research progress if left unaddressed.

**1. Challenge in use case specific evaluation.** Most existing studies in generative model only focus on the fidelity of the synthetic data, i.e. how they resemble the real data in distribution Wang et al. [2019], Tucker et al. [2020], Goncalves et al. [2020], Wang et al. [2021], Kokosi and Harron [2022]. However, additional evaluation is needed to assess the specific use cases. For example, the utility to downstream models and the data privacy. This calls for the introduction of new metrics as well as new evaluation pipelines.

**2. Challenge in off label uses.** Although specialized generative models are developed for one use case, the practical application often requires them to cover multiple use cases (e.g. data augmentation with privacy). Hence, generative models are often used outside the designed scope. Prior work has shown that this may lead to undesirable and unpredictable consequences [Pereira et al., 2021, Ganev et al., 2022]. As a result, researchers need to comprehensively evaluate the generative model across a variety of use cases to assess the risk of off label uses.

**3. Challenge in comparing with a large number of baselines.** In practice, it is often very challenging to systematically compare with a large number of existing baselines because the interfaces (API) of these models are often inconsistent and incompatible (e.g. they may require different formats of input data and conflicting software dependencies). As a result, the researcher usually needs to spend time and effort to harmonize the code rather than focusing on the research question itself.

**4. Challenge in understanding the performance gain.** Generative models are complex systems that involve many components, such as the model architecture, the objective function, and the hyperparameters. These aspects all encode prior assumptions and inductive biases, which would bring unique strengths and weaknesses to the models [Bond-Taylor et al., 2021]. However, it is often difficult to pinpoint the exact component that leads to the performance gain. Most existing studies evaluate the model as a whole and neglect the role of different components.

**Contribution.** In this work, we present Synthcity, an open-source Python library available on pip and GitHub, as a solution to these benchmark challenges. Synthcity offers diverse data modalities and supports various use cases. It provides an extensive set of evaluation metrics for assessing dataset fidelity, privacy, and utility, making it a robust tool for evaluating synthetic data across different applications. With a wide array of state-of-the-art generators and customizable architectures, users can perform consistent comparisons with existing models, gaining insights into performance improvements. Accessible through an intuitive interface, Synthcity facilitates tabular data generation and augmentation, demonstrated through two case studies. Researchers can employ Synthcity for benchmarks and guidance in synthetic data research

## 2   The synthcity library

### 2.1   Overview of the synthcity workflow

Despite the fragmented landscape in synthetic data research, Synthcity implements a unified workflow for benchmark studies. We formalize the process as follows. Let $X$ be the random variable of interest (which could be static, temporal or censored). The real data is composed with observations $x_i \sim P(X)$ drawn from the true (but unknown) distribution. For benchmark evaluation, the real data is split into a training ($\mathbb{D}^r_{train}$) and test ($\mathbb{D}^r_{test}$) set. The generator is trained using the training set $\mathbb{D}^r_{train}$. During training, the generator (explicitly or implicitly) learns the distribution $\hat{P}(X)$ in order to sample from it. After training, the generative model generates synthetic data $\mathbb{D}^s$, which will be evaluated with respect to the test set $\mathbb{D}^r_{test}$ (or in some special cases, the training set $\mathbb{D}^r_{train}$).

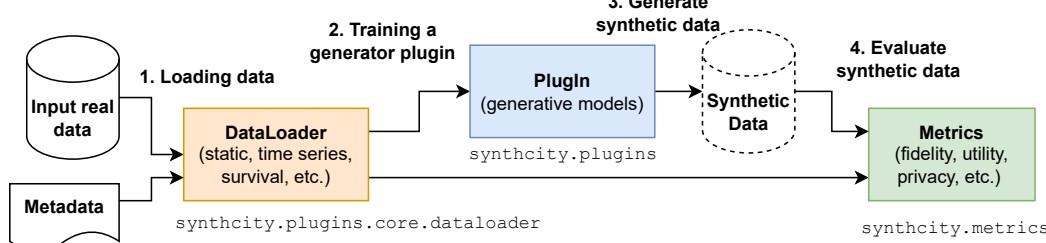

Figure 1: Standard workflow of generating and evaluating synthetic data with synthcity.

The synthcity library captures the entire workflow of synthetic data benchmark in four steps (Figure 1). This workflow applies to all use cases, generative models and data modalities.

1. *Loading the dataset using a DataLoader*. The DataLoader class provides a consistent interface for loading, storing, and splitting different types of input data (e.g. tabular, time series, and survival data). Users can also provide meta-data to inform downstream algorithms, like specifying sensitive columns for privacy-preserving algorithms.

2. *Training the generator using a Plugin*. In synthcity, the users instantiate, train, and apply different data generators via the Plugin class. Each Plugin represents a specific data generation algorithm. The generator can be trained using the fit() method of a Plugin.

3. *Generating synthetic data*. After the Plugin is trained, the user can use the generate() method to generate synthetic data. Some plugins also allow for conditional generation.

4. *Evaluating synthetic data*. Synthcity provides a large set of metrics for evaluating different aspects of synthetic data. The Metrics class allows users to perform evaluation.

In addition, synthcity also has a Benchmark class that wraps around all four steps. This provides a one-line interface for comparing and evaluating different generators and produces an evaluation report at the end of the process.

## 2.2 Evaluation for diverse use cases

Synthetic data has many different use cases including fairness, privacy, data augmentation. As a benchmarking framework, Synthcity provides the users with a comprehensive list of metrics and routines to evaluate various aspects of synthetic data, including metrics that are specific to these use cases. In this section, we describe the use cases of synthetic data and how Synthcity performs its evaluation. A full list of metrics can be found in Appendix.

### 2.2.1 Standard data generation

Standard data generation refers to the most basic generation task, where the synthetic data should be generated as faithfully as possible to the real-data distribution [van Breugel et al., 2023, Hansen et al., 2023]. This is captured by the fidelity metrics.

**Fidelity**. The fidelity of synthetic data captures how much the synthetic data resembles real data. The fidelity metrics typically evaluate the closeness between the true distribution $P$ and the distribution learned by the generator $\hat{P}$ using samples from these two distributions. Synthcity supports distributional divergence measures (e.g. Jensen-Shannon distance, Wasserstein distance, and maximal mean discrepancy) as well as two sample detection scores (i.e. scores that measure how well a classifier can distinguish real versus synthetic data) [Gretton et al., 2012, Lopez-Paz and Oquab, 2016, Snoke et al., 2018].

| Use case | Method | Evaluation | Reference |
|---|---|---|---|
| Standard data generation | Generative model | Fidelity | Gretton et al. [2012] |
| Cross domain augmentation | Domain transfer | Utility | Bing et al. [2022] |
| ML fairness | Balancing distribution | Minority performance | Lu et al. [2018] |
| | Causal fairness | Algorithmic fairness | Xu et al. [2018] |
| Privacy preservation | Differential privacy | Privacy metrics | Abadi et al. [2016] |
| | Threat model | Attack simulation | Shokri et al. [2017] |

Table 1: Synthcity is a unified framework to benchmark diverse use cases of synthetic data. It supports a range of methods and evaluation metrics, and also allows evaluation of off label uses.

### 2.2.2 Cross domain augmentation

Here we consider a dataset that is collected from multiple domains or sources (e.g. data from different countries). Often the practitioner is interested in augmenting one particular data source that suffers from data scarcity issues (e.g. it is difficult to collect data from remote areas) by leveraging other related sources. This challenge has been studied in the deep generative model literature [Antoniou et al., 2017, Dina et al., 2022, Das et al., 2022, Bing et al., 2022]. By learning domain-specific and domain-agnostic representations, the generator is able to transfer knowledge across domains, making data augmentation more efficient. Synthcity offers a clean interface so that the user can benchmark the downstream utility of cross-domain generation using only one line of code..

**Utility.** Synthcity measures the performance for cross domain augmentation through its utility to downstream tasks. Our approach adapts the common practice of train-on-synthetic evaluate-on-real [Beaulieu-Jones et al., 2019], where a downstream predictive model is trained on fully synthetic training data and then validated on real testing data.

For data augmentation, Synthcity augments the data-scarce domain in the training data $\mathbb{D}^r_{train}$ with the synthetic data $\mathbb{D}^s$. A predictive model is then trained on this augmented dataset and evaluated on the domain of interest in the testing data $\mathbb{D}^r_{test}$. Synthcity supports various types of predictive tasks, including regression, classification and survival analysis. In addition to linear predictive models, synthcity supports Xgboost and neural nets as downstream models due to their wide adoption in data analytics. In practice, the user may average the performance of several predictive models to reduce the model uncertainty.

As a naive baseline, Synthcity reports the predictive performance where no data augmentation is performed. Synthcity provides a pre-configured pipeline to automatically handle this entire procedure, reducing the code and preventing mistakes.

### 2.2.3 Synthetic data for ML fairness

Existing research has considered two different ways where Synthetic data could promote fairness. Table 2 shows the corresponding models in synthcity.

*1. Balancing distribution.* In this setting, certain groups of people are underrepresented in a dataset that is used for training downstream ML systems. This may lead to a bias being introduced into these ML systems [Lu et al., 2018, de Vassimon Manela et al., 2021, Kadambi, 2021]. As a remedy, one could generate synthetic records for the minority group to augment the real data, thereby achieving balance in distribution. This often requires the data generator to learn the conditional distribution $P(X|G)$, where $G$ is the group label.

*2. Causal fairness.* The second approach is to generate fairer synthetic data from a biased real dataset and to use synthetic data alone in downstream tasks [Zemel et al., 2013, Xu et al., 2018, 2019a, van Breugel et al., 2021]. In this setting, it is postulated that the real distribution $P(X)$ reflects existing biases (e.g. unequal access to healthcare). The task for the generator is to learn a distribution $\hat{P}(X)$ that is free from such biases but also stay as close to $P(X)$ as possible (to ensure high data fidelity). Typically, notions of causality are employed in the bias removal process.

**Fairness.** Synthcity allows users to benchmark both use cases by training a downstream predictive model on the fully synthetic or augmented data and presenting their performance or characteristics. For example, one can evaluate the performance gain on any specified (minority) group as an indicator of the utility of synthetic data. In addition, Synthcity also supports standard algorithmic fairness metrics for the trained predictive model, such as Fairness Through Unawareness, Demographic Parity and Conditional Fairness [van Breugel et al., 2021]

### 2.2.4 Synthetic data for privacy

Methods for generating privacy-preserving synthetic data mainly fall into two categories: the ones that employ differential privacy, and the ones that are designed for specific threat models.

*1. Differential privacy (DP).* DP is a formal way to describe how private a data generator is [Dwork, 2008]. Typically, generators with DP property introduce additional noise in the training procedure [Jordon et al., 2022]. For example, adding noise in the gradient or using a noisy discriminator in a GAN architecture [Abadi et al., 2016, Jordon et al., 2018, Long et al., 2019].

*2. Threat model (TM).* While DP focuses on giving formal guarantees, the TM approach is designed for specific threat models, such as membership inference, attribute inference, and re-identification [Shokri et al., 2017, Kosinski et al., 2013, Dinur and Nissim, 2003]. These models often involve regularization terms designed to mitigate privacy attack risk [Yoon et al., 2020].

**Privacy.** Synthcity evaluates the privacy of synthetic data using a list of well-established privacy metrics (e.g. k-anonymity [Sweeney, 2002] and l-diversity [Machanavajjhala et al., 2007]). Furthermore, it can measure the privacy of data by performing simulated privacy attacks (e.g. a re-identification attack). The success (or failure) of such an attack quantifies the degree of privacy preservation.

### 2.2.5 Evaluating off label use cases

Synthcity allows users to conveniently evaluate the off label usage of generative models. For instance, one could evaluate the privacy of synthetic data even if they are not generated by a privacy-enabled generative model. As another example, one could evaluate the fairness for generative models that are differentially private, thereby enabling studies like Ganev et al. [2022].

Off-label evaluation is made easy because Synthcity implements generative models and evaluation metrics in two separate modules (PlugIns and Metrics). The consistent interface enables mix and match of models and metrics to empower different benchmark studies.

### 2.3 Baseline generative models

As a benchmarking framework, Synthcity is a one-stop-shop for state-of-the-art benchmarks with a large collection of baselines covering both deep generative models and other types of generative models. In this way, the user can easily compare with a range of existing methods, without the need to worry about implementation details or interfaces. Table 2 lists the generative models in synthcity for different data modalities.

Synthcity covers all major families of deep generative models, including Generative adversarial networks (GAN) [Goodfellow et al., 2020], Variational Autoencoders (VAE) [Kingma et al., 2019], Normalizing flows (NF) [Papamakarios et al., 2021], as well as Diffusion models (DDPM) [Kingma et al., 2021]. In the GAN family, Synthcity currently supports GOGGLE [Liu et al., 2023], CTGAN Xu et al. [2019b], DPGAN [Xie et al., 2018], PATEGAN [Jordon et al., 2019], ADSGAN [Yoon et al., 2020], DECAF [van Breugel et al., 2021] for static data, Survival GAN [Norcliffe et al., 2023] for censored data, TimeGAN [Yoon et al., 2019] for time series data, as well as RadialGAN [Yoon et al., 2018] for multi-source data. In the VAE family, it supports TVAE [Xu et al., 2019b], RTVAE for static data [Akrami et al., 2020], Survival VAE [Norcliffe et al., 2023] for censored data, and TimeVAE [Yoon et al., 2019] for time series data. In the NF family, Synthcity implements the standard NF [Papamakarios et al., 2021] for static data, Survival NF [Norcliffe et al., 2023] for censored data, and FourierFlow [Alaa et al., 2021] of time series data. Synthcity also includes the TabDDPM

| Data Modality | Model | Standard Gen | Privacy | | Fairness | |
|---|---|---|---|---|---|---|
| | | | DP | TM | Balance | Causal |
| Static | Bayesian Net | √ | | | | |
| | NF | √ | | | | |
| | GREAT | √ | | | | |
| | ARF | √ | | | | |
| | GOGGLE | √ | | | | |
| | TabDDPM | √ | | | | |
| | TVAE | √ | | | √ | |
| | RTVAE | √ | | | √ | |
| | CTGAN | √ | | | √ | |
| | AIM | √ | √ | | | |
| | PrivBayes | √ | √ | | | |
| | DPGAN | √ | √ | | | |
| | PATEGAN | √ | √ | | | |
| | ADSGAN | √ | | √ | | |
| | DECAF | √ | | | | √ |
| Static (Censored) | Survival GAN | √ | | √ | √ | |
| | Survival VAE | √ | | | | |
| | Survival NF | √ | | | | |
| Time Series (regular, irregular, censored) | TimeGAN | √ | | | √ | |
| | TimeVAE | √ | | | | |
| | FourierFlow* | √ | | | | |
| | Probabilistic AR* | √ | | | | |
| Multi-source | RadialGAN | √ | | | √ | |

Table 2: Generative models available in synthcity for different data modalities and use cases. Abbreviations: Differential Privacy (DP), Threat Model (TM). *FourierFlow and Probabilistic AR is compatible with regular time series only while TimeGAN and TimeVAE support both.

[Kotelnikov et al., 2022] in the diffusion model family, and GREAT Borisov et al. [2022], which uses auto-regressive generative LLM model.

In addition to deep generative models, Synthcity also contains generative models that are not based on neural networks, such as Bayesian networks [Heckerman, 1997], AIM [McKenna et al., 2022], Probabilistic Auto-regressive models [Deodatis and Shinozuka, 1988] and Adversarial random forests (ARF) [Watson et al., 2023].

Synthcity implements all generative models using the PlugIn interface. This consistent approach makes it easy to add additional generative models into the benchmark. The GitHub repository includes tutorials and step-by-step instructions on how to add new models.

## 2.4 Architecture and hyper-parameters

To help researchers pinpoint the source of performance gain and conduct fair comparison, Synthcity allows the user to incarnate all the deep generative models with different network architectures and hyper-parameters.

The architecture can be specified when the user creates a model instance (i.e. a PlugIn). For example, each time-series generative model can be configured using twelve different architectures, including LSTM [Hochreiter and Schmidhuber, 1997], GRU [Dey and Salem, 2017], Transformer [Vaswani et al., 2017], MLSTM-FCN [Karim et al., 2019], TCN [Lea et al., 2017], InceptionTime and InceptionTimePlus [Ismail Fawaz et al., 2020], XceptionTime [Rahimian et al., 2020], ResCNN [Sun et al., 2020], Omni-Scale CNN [Tang et al., 2020], and XCM [Fauvel et al., 2021]. The network architectures compatible with other data modalities are tabulated in the Appendix.

Synthcity also has a consistent interface for dealing with hyper-parameters. The library allows the user to list, set, and sample all relevant hyper-parameters of a generative model. Furthermore, this interface is compatible with all popular hyper-parameter optimization libraries, such as Optuna [Akiba et al., 2019]. In this way, Synthcity allows the user to perform hyper-parameter search before evaluating on the best-performing setting to ensure a like-for-like comparison. Furthermore, Synthcity also allows the user to configure various early stopping rules to control and compare the training of generative models.

## 2.5 Data modalities

We emphasize that "tabular data" in fact encapsulates many different data modalities, including static tabular data, time series data, and censored survival data, all of which may contain a mix of continuous and discrete features (columns). Synthcity can also handle composite datasets composed of multiple subsets of data. We give a detailed description of the diverse tabular data modalities Synthcity supports in Figure 2 and further discuss them below. In future versions, we plan to include more data modalities including relational database-style data, richly annotated images, and texts.

### 2.5.1 Single dataset

We start by introducing the most fundamental case where there is a single training dataset (e.g. a single DataFrame in Pandas). We characterize the data modalities by two axes: the *observation pattern* and the *feature type*. Synthcity supports all combinations.

The observation pattern describes whether and how the data are collected over time. There are three most prominent patterns, static data, regular time series, and irregular time series, which are all supported by synthcity.

The second axis, feature type, describes the domain of individual features. Synthcity supports multivariate tabular data with mixtures of continuous, categorical, integer, and censored features. Censored features are common in survival analysis applications (e.g. healthcare and insurance). They are represented as a tuple $(x, c)$, where $x \in \mathbb{R}^+$ represents the survival time and $c \in \{0, 1\}$ is the censoring indicator.

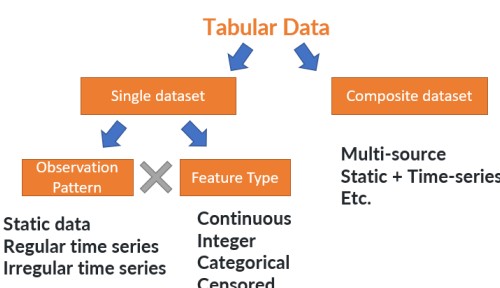

Figure 2: Supported tabular data modalities.

### 2.5.2 Composite dataset

A composite dataset involves multiple sub datasets. Synthcity can handle the benchmarking of different classes of composite datasets. Currently, it supports (1) static datasets with the same features, collected from different domains, (2) a static and a time series dataset. The latter setting is common in applications. For example, a patient's medical record may contain both static demographic information and longitudinal follow up data.

## 3 Comparison with existing libraries

In this section, we compare synthcity with other popular open source libraries for synthetic data generation to demonstrate its suitability as a comprehensive benchmark framework. Here we only consider the libraries that can generate synthetic data while preserving the statistical properties of real data, which includes YData Synthetic, Gretel Synthetics, SDV, DataSynthesizer, SmartNoise and nbsynthetic. Libraries that generate "fake" data for software testing are not considered because they do not attempt to learn the distribution of real data.

| Setting \ Software | Synthcity | YData | Gretel | SDV | DataSynthesizer | SmartNoise | nbsynthetic |
|---|---|---|---|---|---|---|---|
| **Data modalities** | | | | | | | |
| Static data | ✓ | ✓ | ✓ | ✓ | ✓ | ✓ | ✓ |
| Regular time series | ✓ | ✓ | ✓ | ✓ | | | |
| Irregular time series | ✓ | | | | | | |
| Censored features | ✓ | | | | | | |
| Composite data | ✓ | | | ✓ | | | |
| **Use cases** | | | | | | | |
| Generation | ✓ | ✓ | ✓ | ✓ | ✓ | ✓ | ✓ |
| Fairness (balance) | ✓ | ✓ | ✓ | ✓ | | | ✓ |
| Fairness (causal) | ✓ | | | | | | |
| Privacy (DP) | ✓ | | ✓ | | | ✓ | |
| Privacy (TM) | ✓ | | | | | | |
| Cross domain aug. | ✓ | | | | | | |

Table 3: The data modalities and use cases supported by synthcity and other open source synthetic data libraries. Comparisons are based on the software versions available at the time of writing.

Table 3 shows that synthcity supports much broader use cases and data modalities than the alternatives. The existing libraries often focus on a single data modality or use case because they are intended as a solution to a specific problem rather than a benchmark framework. Furthermore, Synthcity includes many more data generators, including all major flavors of deep generative models as well as traditional generative models. It also contains a built-in evaluation module that assesses various aspects of the generator. A more detailed comparison of the supported data generators and evaluation metrics are available in the Appendix. The broad coverage of data modalities, use cases, data generators and evaluation metrics make Synthcity unique in its capacity for model evaluation and comparison.

## 4  Illustrative case studies

In this section, we present two illustrative use cases to show the type of benchmark studies that Synthcity can facilitate. We stress that these examples do not cover the full capability of Synthcity and they are used as illustrations.

### 4.1  Static tabular data generative model benchmark

We study which generative model has the strongest performance in generating synthetic tabular data. Synthcity allows us to compare a variety of state-of-the-art algorithms in this study, including ARF, GOGGLE, TabDDPM, CTGAN and TVAE. These algorithms are representative of broader families of generative models such as GANs, VAEs, Diffusion models, and forest-based generative models.

Similar to prior tabular data benchmarks [Grinsztajn et al., 2022], we have selected 18 datasets from the OpenML benchmark, which cover common regression and classification datasets encountered in data science projects [Vanschoren et al., 2014]. The datasets cover a range of sample sizes (4,209 to 1,025,010) and feature counts (5 to 771).

Synthcity can automatically calculate more than 25 supported evaluation metrics in a benchmark. In this study, we focus on evaluating the fidelity of synthetic data. Similar to Liu et al. [2023], we report the average of the three-dimensional metrics ($\alpha$-precision, $\beta$-recall, and authenticity), as proposed in Alaa et al. [2022], as a measure of data quality—whether the synthetic data are realistic, cover the true data distribution, and are generalized. Furthermore, we report the detection score, which reflects how often the synthetic data can be distinguished from the real data. To reduce the variability from the classifiers, we report the average AUROC scores from three different post-hoc data classifiers, as in Liu et al. [2023].

Table 4 shows the experimental results averaged across all the datasets. We observe that the ARF model achieves strong performance in the quality score and stands out in terms of the detection score.

This suggests that the tree-based generative models are strong competitors to deep generative models for static tabular data. And this area is a promising avenue for further research.

|  | Quality | Detection |
|---|---|---|
| ARF | 0.5475 | 0.6721 |
| GOGGLE | 0.4054 | 0.9261 |
| TabDDPM | 0.5436 | 0.7074 |
| CTGAN | 0.5475 | 0.7758 |
| TVAE | 0.5487 | 0.7389 |

Table 4: Benchmark results for static tabular data generation. Quality: the higher the better; Detection: the lower the better.

## 4.2 Tabular data fairness and augmentation benchmark

We consider a benchmark study on cross-domain data augmentation for improving predictive performance on minority groups. We use the SIVEP-Gripe public dataset as an illustrative example, which contains anonymized records of COVID-19 patients in Brazil [Baqui et al., 2021]. In this dataset, 'Mixed' and 'White' are the majority ethnicity groups while 'Black', 'East Asian' and 'Indigenous' are the minority groups (accounting for less than 10% of the total population). The dataset is used for training a downstream model to predict COVID-19 mortality. Due to the distributional imbalance, the downstream predictor is likely to under-perform on the minority groups, which may raise fairness issues (Section 2.2.3). This study aims to benchmark the utility of different generative models for data augmentation by measuring the AUROC of mortality prediction on the minority groups.

Synthcity allows us to easily compare RadialGAN, which was designed for cross-domain data augmentation, and the conditional generative models (TabDDPM, CTGAN, and TVAE). We use Synthcity's pre-configured pipeline for data augmentation benchmark, which reduces the amount of code and prevents data leakage. Synthcity also allows us to evaluate the performance gain for different downstream models, and we have selected a multi-layer perceptron classifier and a xgBoost classifier. The results are listed in Table 5. We observe that data augmentation consistently improves the accuracy of mortality prediction for minority groups. TabDDPM, a novel diffusion model, achieves the best overall performance, followed by RadialGAN.

|  | Neural net | XgBoost |
|---|---|---|
| TabDDPM | 0.7241 | 0.7786 |
| RadialGAN | 0.7137 | 0.7627 |
| CTGAN | 0.6477 | 0.7507 |
| TVAE | 0.3623 | 0.7794 |
| Baseline | 0.3244 | 0.7327 |

Table 5: Benchmark results for cross-domain data augmentation. The metric reported is the AUROC of mortality prediction on the minority groups

## 5 Discussion

Synthetic data is an emerging field where many novel algorithms have been proposed; yet there lacks an easy way to benchmark generative models across different desired or off label use cases, compare them with diverse baselines, and explain their performance gain. In this work, we present the open source Synthcity library as a solution to the benchmark challenge. Synthcity contains many built-in generative models, architectures and evaluation metrics, which are easily accessible through end-to-end evaluation pipelines. It can help researchers to perform in-depth and comprehensive benchmark studies with minimal programming effort.

# 6  Funding Statement

This work is supported by Cancer Research UK. The computational experiments in this work was supported by Azure sponsorship credits granted by Microsoft's AI for Good Research Lab.

# 7  Acknowledgment

We thank all the open source contributors to the Synthcity library. The full list of contributors is available on GitHub. Special thanks goes to Bogdan Cebere, who made major contributions to Synthcity while he was working as a Research Engineer at the van der Schaar Lab.

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

# A  Appendix

## A.1  Code availability

The code for the illustrative use cases are available on GitHub https://github.com/vanderschaarlab/synthcity-benchmarking. The synthcity library is available on pip and GitHub. The tutorials folder contains additional illustrative examples.

## A.2  Supported algorithms and metrics

| Aspect | Evaluation Metric \Software | Synthcity | YData | Gretel | SDV | DataSynthesizer | SmartNoise | nbsynthetic |
|---|---|---|---|---|---|---|---|---|
| Fedelity | Jensen-Shannon distance | √ | | | | | | |
| | Wasserstein distance | √ | | | | | | |
| | Total variation distance | | | | √ | | | |
| | KL divergence | √ | | | | | | |
| | Skewness | | | | √ | | | |
| | Max-mean discrepancy | √ | | | | | | √ |
| | KS test | √ | | | √ | | | √ |
| | PRDC | √ | | | | | | |
| | Alpha–precision | √ | | | | | | |
| | Survival Kaplan-Meier dist. | √ | | | | | | |
| | Detection: linear | √ | | | √ | | | |
| | Detection: NN | √ | | | | | | |
| | Detection: XGB | √ | | | | | | |
| | Detection: Linear | | | | √ | | | |
| Utility | Linear model | √ | | | √ | | | |
| | MLP | √ | | | √ | | | |
| | XGBoost | √ | | | √ | | | |
| | Static survival | √ | | | | | | |
| | Time-series | √ | | | | | | |
| | Survival time-series | √ | | | | | | |
| Privacy | Correct attribution prob. | √ | | | √ | | | |
| | K-anonymity | √ | | | | | | |
| | K-map | √ | | | | | | |
| | Delta-presence | √ | | | | | | |
| | L-diversity | √ | | | | | | |
| | DOMIAS | √ | | | | | | |
| | Identifiability score | √ | | | | | | |

Table 6: The evaluation metrics supported by synthcity and other open source synthetic data libraries. Comparisons are based on the software versions available at the time of writing.

| Static | Censored | Time Series |
|---|---|---|
| Fully connected | Weibull AFT | LSTM |
| Residual network | Cox PH | GRU |
| TabNet | Random Survival Forest | RNN |
| | Survival Xgboost | Transformer |
| | Deephit | MLSTM_FCN |
| | Tenn | TCN |
| | Date | InceptionTime |
| | | InceptionTimePlus |
| | | XceptionTime |
| | | ResCNN |
| | | OmniScaleCNN |
| | | XCM |

Table 7: Available network architectures and survival models in synthcity for different data modalities. These components are compatible with multiple algorithms.

| Algorithm \Software | Synthcity | YData | Gretel | SDV | DataSynthesizer | SmartNoise | nbsynthetic |
|---|---|---|---|---|---|---|---|
| AIM | √ | | | | | | |
| GREAT | √ | | | | | | |
| TabDDPM | √ | | | | | | |
| ARF | √ | | | | | | |
| GOGGLE | √ | | | | | | |
| CTGAN | √ | √ | | √ | | | √ |
| ACTGAN | | | √ | | | | |
| TVAE | √ | | | √ | | | |
| Bayesian Network | √ | | | | | | |
| Normalizing Flows | √ | | | | | | |
| Survial GAN | √ | | | | | | |
| Survival VAE | √ | | | | | | |
| DoppelGANger | | | √ | | | | |
| TimeGAN | √ | √ | | | | | |
| FourierFlows | √ | | | | | | |
| Probabilistic AR | √ | | | √ | | | |
| DECAF | √ | | | | | | |
| RadialGAN | √ | | | | | | |
| ADSGAN | √ | | | | | | |
| DPGAN | √ | | √ | | | √ | |
| PATEGAN | √ | | | | | √ | |
| PrivBayes | √ | | | | √ | | |

Table 8: The data generating algorithms supported by synthcity and other open source synthetic data libraries. Comparisons are based on the software versions available at the time of writing.

