# OpenReview forum: "Synthcity: a benchmark framework for diverse use cases of tabular synthetic data"
_NeurIPS.cc/2023/Track/Datasets_and_Benchmarks — NeurIPS 2023 Datasets and Benchmarks Poster_

### Official Review · Reviewer_rLVX · 2023-07-17
**Synthcity is comprehensive and well documented, but need to discuss some of the state-of-the-art that are not included**

**Rating:** 7
**Confidence:** 4
**Clarity:** Yes, the paper is well written, it's …

**Strengths:**

The GitHub documentation of Synthcity is comprehensive, offering a wealth of resources including tutorials that facilitate quick learning. The library's APIs are user-friendly, allowing for effortless integration of new algorithms. Notably, the benchmark encompasses a diverse range of algorithms that are tailored to handle various data types such as tabular data, time series data, and images. Additionally, the benchmark provides an extensive array of metrics, enabling thorough comparisons of model performance with regard to privacy and fairness considerations. Overall, the strength of this library lies in its accessibility, versatility, and its ability to foster comprehensive evaluations and comparisons of synthetic data generation models.

**Additional Feedback:**

None.

**Correctness:**

Yes, the evaluation methods and experiment design are appropriate and performed correctly.

**Documentation:**

Yes, a GitHub repo is provided, and the documentation is clear and comprehensive

**Ethics:**

No.

**Limitations:**

See "Opportunities For Improvement", the paper didn't include some of the state-of-the-art approaches.

**Opportunities For Improvement:**

In the paper “Benchmarking Differentially Private Synthetic Data Generation Algorithms”, it’s shown that Marginal-based methods consistently outperformed other methods, and GAN-based methods were unable to preserve the 1- dimensional statistics of tabular data.

It is worth noting that the PrivBayes method, while once considered state-of-the-art, has since been surpassed by more advanced techniques. Presently, the leading methods in differentially private synthetic data generation include,
1. "AIM: An Adaptive and Iterative Mechanism for Differentially Private Synthetic Data" for categorical attributes
2. "Private Synthetic Data for Multitask Learning and Marginal Queries" for numerical attributes.

Notably, the current benchmark does not encompass these state-of-the-art approaches, indicating a need for their inclusion in DP-related synthetic data generators for comprehensive evaluations.


**Relation To Prior Work:**

Yes, it is compared with previous benchmarks.

**Summary And Contributions:**

In this study, the authors introduced a novel Python library called Synthcity, which addresses various challenges encountered in benchmarking synthetic data generators. The library not only offers use case-specific evaluation capabilities but also facilitates comparisons with a wide range of baselines and facilitates an understanding of performance improvements. Furthermore, Synthcity includes a comprehensive set of evaluation metrics and offers access to a diverse collection of state-of-the-art generators.

---

> ### Author Response · Authors · 2023-08-15
> **Reply to rLVX**
>
> Dear Reviewer,
>
> We appreciate your insightful feedback and comments. We have updated the library and the paper to address all of your concerns.
>
> ### SOTA DP models
>
> Thanks for pointing out additional SOTA DP models. We will cite and refer to these works both in the paper and in the library.
> We are committed to maintain and update Synthcity in this fast-moving field to include new SOTA methods.
>
> 1. AIM. We have added AIM in Synthcity (see this [pull request](https://github.com/vanderschaarlab/synthcity/pull/216)). We have updated the manuscript accordingly. Thank you.
> 2. "Private Synthetic Data for Multitask Learning and Marginal Queries". Unfortunately, there is no publicly available code for this paper. We have contacted the authors, who have kindly replied and stated that the official code would be released soon. We will start to integrate the method as soon as the official code is released.
>
> Thank you!

---

> > ### Comment · Reviewer_rLVX · 2023-08-17
> >
> > The authors have addressed my concerns, I'm willing to update my review to accept.

---

> > > ### Author Response · Authors · 2023-08-17
> > >
> > > We are really glad that our answers have addressed your concerns. We also much appreciate that you are willing to increase the score.

---

### Official Review · Reviewer_qioZ · 2023-07-24
**Review for Synthcity**

**Rating:** 6
**Confidence:** 4
**Clarity:** The paper is clearly written.

**Strengths:**

- **Diverse use cases**: The library covers a variety of different use-cases from evaluating synthetic data quality to privacy evaluations of generated data.
- **Reasonable library structure**: The library structure 'load, train, generate, evaluate' appears to be reasonable. I like that the authors allow for meta-data to be supplied for downstream evaluations. How difficult is it for other researchers to include new models and/or evaluation measures?


**Additional Feedback:**

- In the attack evaluation section, the paper mentions "attack simulation" and references Shokri et al. (2017) and "privacy metrics." However, the text does not elaborate on the specific attacks or metrics that the library incorporates, leaving readers with an unclear understanding of this aspect.

- How difficult is it for other researchers to include new models and/or evaluation measures?

**Correctness:**

To evaluate this aspect more clearly, it would have been helpful if code usage examples would have been given. The overall library structure seems reasonable (please see my comments in 'strenghts' and 'opportunities for improvement' above).

**Documentation:**

Yes, there exists a link to the auhtors' github repository.

**Ethics:**

No ethical concerns.

**Limitations:**

Authors addressed this point.

**Opportunities For Improvement:**

**SOTA metrics and models**: Several state-of-the-art tabular data generation models and standard metrics from generation literature would improve the scope of the library. Moreover, important standard metrics from privacy literature to evaluate the efficacy of membership inference are missing:
- Data generation metrics:
  - Machine learning efficiency metric: The generated data set should replace the real data in a training process and hence this measure evaluates the performance of discriminative models trained on synthetic data sets.
  - Metric that quantifies whether copying of data happens (e.g., distance to closest records): To verify that the generated data is similar to original samples while not being exact copies.

- State-of-the-art tabular data generation models:
  - GREAT uses an auto-regressive generative LLM model to generate tabular data (Borisov et al (2022), https://openreview.net/forum?id=cEygmQNOeI)
  - STASY uses a score-based generative model to generate tabular data (Kim et al (2023), https://openreview.net/forum?id=1mNssCWt_v)
- Membership inference attacks:
  - LIRA (Likelihood ratio attack): This is a likelihood-ratio based attack that is built on the Neyman-Pearson lemma which constitutes the state-of-the-art attack to assess privacy risks (Carlini et al (2021), https://arxiv.org/abs/2112.03570).
  - For completeness, very recently, a stronger likelihood attack was proposed that uses gradients as opposed to model confidence scores (Leemann  et al (2023), https://arxiv.org/abs/2306.07273).
- DP privacy evaluations:
  - Adversary instantiation: To understand the importance of different adversary capabilities, it is essential to feature a variety of privacy attacks that reflect different capabilities including the worst-case attack on DP private models (Nasr et al (2021), https://arxiv.org/pdf/2101.04535).

**No usage examples**: There is an absence of usage examples in the paper. Providing code usage examples would have been beneficial for users, illustrating how to load data and compute metrics using the library effectively.



**Relation To Prior Work:**

Relation to prior work is mostly discussed. Please refer to comments in 'opportunities for improvement' above for suggestions of works that should be included into the library.

**Summary And Contributions:**

"Synthcity" is an open-source Python library designed to facilitate fast benchmarking of synthetic data generators for researchers and practitioners. The strength of this framework lies in its ability to cater to diverse use cases, encompassing evaluations of synthetic data quality, fairness issues and privacy assessments of generated data. The library seems to be well-organized, featuring loading, training, generating, and evaluating data which could allow for seamless integration of new models and evaluation measures by other researchers.
The paper's major strength lies in its coverage of various use cases, accommodating the evaluation of synthetic data quality and privacy measures for generated data.
However, the paper does have some notable weaknesses: First, it lacks state-of-the-art (SOTA) data generation metrics metrics and models. Second the absence of important standard metrics from the privacy literature, like Membership Inference using LIRA (Likelihood ratio attack), and worst-case DP attack evaluations hinder a comprehensive evaluation of privacy risks associated with the generated data.
Given the 9-page limit, carefully incorporating these improvements will undoubtedly elevate the quality and relevance of the paper.

---

> ### Author Response · Authors · 2023-08-15
> **Reply to qioZ**
>
> Dear Reviewer,
>
> We appreciate your thorough feedback and comments. We have updated the library and the paper to address all your concerns.
>
> ### SOTA metrics and models
>
> Thanks for pointing out additional SOTA models and metrics. We will cite and refer to these works both in the paper and in the library. We provide a point-by-point response below.
>
> 1. Data generation metrics. Synthcity already included the two metrics you pointed out. We will clarify this in the paper.
> 	- Machine learning efficiency metric. The `PerformanceEvaluator` performs train-on-synthetic test-on-real evaluation to measure the utility of synthetic data to a downstream supervised learning task [(Github link)](https://github.com/vanderschaarlab/synthcity/blob/7172a46e046616972845b23beec62fb9618037f5/src/synthcity/metrics/eval_performance.py#L48).
> 	- Copying of data. The `NearestSyntheticNeighborDistance` computes the average distance from the real data to its closest neighbour in the synthetic data (zero means exact copying) [(Github link)](https://github.com/vanderschaarlab/synthcity/blob/7172a46e046616972845b23beec62fb9618037f5/src/synthcity/metrics/eval_sanity.py#L174).
> 2. SOTA tabular data generator
> 	- We have added GREAT in Synthcity (see this [pull request](https://github.com/vanderschaarlab/synthcity/pull/214)) and updated the manuscript accordingly.
> 	- We have created an [issue](https://github.com/vanderschaarlab/synthcity/issues/219) on GitHub for the integration of STASY and we're currently working on it. Synthcity already includes TabDDPM, which is a score-based diffusion model for tabular data.
> 3. Membership inference attacks (MIA). To date, most of the MIA works focus on attacks on a discriminative model, including the two works you have referred to. In contrast, Synthcity considers data generation where MIA aims to infer whether a test point $x$ is in the training set of a *generative model*.
> 	-  Specifically, LIRA makes use of the discriminative loss function (i.e. cross entropy loss) Whether the procedure generalizes to generative losses (e.g ELBO or even the GAN loss) is beyond the scope of the current paper.
> 	-  Synthcity has included a recently proposed MIA in the generative setting (DOMIAS), which is also based on the likelihood ratio. We will keep updating the library to include the SOTA methods in this area. Thank you.
> 4. DP privacy evaluations. We have created an [issue](https://github.com/vanderschaarlab/synthcity/issues/220) on GitHub for the integration of Adversary Instantiation and it will be included in the next major update and we will cite this work in the paper.
>
> ### Usage examples
>
> Due to the page limit, we did not manage to include usage examples in the main paper. However, the library itself contains comprehensive tutorials, covering all functionalities [(Github link)](https://github.com/vanderschaarlab/synthcity/tree/main/tutorials).
>
> ### Adding new models or metrics
>  We agree with the reviewer. Synthcity's modular structure makes it easy to include additional models and metrics. We have made tutorials and guidelines for researchers to add their own models and metrics [(Tutorial link)](https://github.com/vanderschaarlab/synthcity/blob/main/tutorials/tutorial1_add_a_new_plugin.ipynb) [(Guideline link)](https://github.com/vanderschaarlab/synthcity/blob/main/CONTRIBUTING.MD).  In fact, the recent inclusion of ARF and TabDDPM benefited from the community contribution. Currently, 7 out of 11 contributors to the library are from the open source community.
>
>  Thank you.

---

> > ### Comment · Reviewer_qioZ · 2023-08-17
> > **Response to author rebuttal**
> >
> > Thank you for your response. Most of my concerns have been addressed.

---

### Official Review · Reviewer_T6Ey · 2023-07-24
**A promising all-encompassing benchmark with minor flaws and lack of sufficient evaluation**

**Rating:** 6
**Confidence:** 4

**Strengths:**

The main strength of this work is the vastness of functionality it supports such as domain augmentation, privacy, fairness etc.
I am not aware of any prior works that support multiple such modalities in an API-like format. Therefore, if done well, Synthcity could foster research progress in synthetic data generation.
The authors have done a great job by including tutorials that enable contributors or users to get onboarded quickly.

**Additional Feedback:**

None

**Clarity:**

The paper can benefit from better organization and more concise re-writing. I have included some of my criticisms in the Limitations section.

**Correctness:**

The contribution of this work is a library and therefore, there is not much to claim except the functionalities offered. The authors have performed a few "Illustrative case studies" which I believe could have been a possible benchmark. However, those case-studies do not appear to be well-structured and probably can benefit from a more comprehensive evaluation.

**Documentation:**

Except for the first page, the documentation is not usable at all. It contains images of object schemas instead of the actual code. I believe this is something that needs a drastic improvement for this library to be useful and attract open-source contributors.

**Limitations:**

Library -
1. I tried two use cases (time series and tabular data generation) provided on their GitHub examples and despite following their code as is, I ran into errors and could not use the library for training models. Therefore, my first impression is that the library is not stable.
2. The documentation is not well-written except for the first page and the GitHub README file.

Writing -
1. The paper can be significantly improved by organizing it in a more structured manner and improving the writing. To highlight a few issues -
At L65: "to assess various aspects" is too vague and could mean anything. The contribution paragraph generally could be improved by quantifying contributions concretely.
At L123: "Synthcity can benchmark this mode of cross-domain generation via the synthetic data’s utility to downstream tasks." Why is that special to synthcity? Anyone can evaluate any benchmark by using synthetic data's utility for downstream tasks
At L138: "Preventing mistakes such as data leakage" I am not sure why preventing data leakage becomes particularly relevant for cross domain generation?

Illustrative case studies do not appear to be comprehensive enough. The supplementary shows support for a lot of different models and techniques however, none of them have been evaluated in the paper. The same holds true for privacy benchmarking.

**Opportunities For Improvement:**

While the work supports multiple functionalities, it could benefit from more clear documentation on how to use all those models and evaluation metrics.

**Relation To Prior Work:**

The authors compare their library with other alternatives for synthetic data in Table 3. However, it is strange that Synthcity is a superset of all existing synthetic data generation libraries. A more fair comparison probably would talk about functionalities missing in Synthcity but present in others.

**Summary And Contributions:**

The authors propose a library, Synthcity, for synthetic data generation. Their goal is to unify several fragmented extensions and offshoots of synthetic data generation use-case within a single framework to facilitate comprehensive evaluation and enable off-label use cases. Their contribution includes a suite of metrics, models, boilerplate code, and APIs to use them in a plug-and-play manner.

---

> ### Author Response · Authors · 2023-08-15
> **Reply to T6Ey**
>
> Dear Reviewer,
>
> We appreciate your thorough feedback and comments. We have updated the library and the paper to address all your concerns.
>
> ### Example use cases
>
> Thanks for reporting this issue. We have gone through all the example use cases and ensured they run as expected (see this [pull request](https://github.com/vanderschaarlab/synthcity/pull/210)).
> We have also implemented an automated testing mechanism to ensure future software updates will not break these examples.
> All examples are also runnable on Google Colab [(link)](https://github.com/vanderschaarlab/synthcity#-tutorials), a preconfigured ML platform that avoids GPU set-up issues. To run the tutorial, click the `Open in Colab` button, and run the first cell containing `pip install`. After installation completes, restart the runtime (Runtime > Restart the runtime) and run from the second cell containing `import …`.
>
> ### Documentation
>
> We have updated the [documentation page](https://synthcity.readthedocs.io/en/latest/?badge=latest) to cover information about function signature and example usage (see this [pull request](https://github.com/vanderschaarlab/synthcity/pull/211)). This is now available for all user-facing functions and classes. There was a configuration error that caused some information not displayed properly - this has now been fixed. Thank you.
>
> ### Additional use cases
>
> To further illustrate the utility and flexibility of the library, we have included an additional case study that compares the performance of various *neural architectures* for generating synthetic time series. In this example, we have used the [PBC time series dataset](https://stat.ethz.ch/R-manual/R-patched/library/survival/html/pbcseq.html). The results are reported in the table below, which shows that the XceptionTime architecture achieves the best performance on the dataset based on the distributional distance measure and the downstream utility.
>
> |                   | Wasserstein Distance | Downstream Brier Score |
> |-------------------|----------------------|------------------------|
> | LSTM              | 6.55                 | 0.37                   |
> | GRU               | 6.44                 | 0.25                   |
> | RNN               | 6.28                 | 0.34                   |
> | Transformer       | 6.17                 | 0.51                   |
> | MLSTM             | 6.41                 | 0.24                   |
> | InceptionTime     | 6.21                 | 0.25                   |
> | InceptionTimePlus | 6.57                 | 0.35                   |
> | XceptionTime      | **6.06**             | **0.22**               |
> | ResCNN            | 6.48                 | 0.37                   |
> | OmniScaleCNN      | 6.52                 | 1.00                   |
> | XCM               | 6.79                 | 1.00                   |
>
> We would like to clarify further that the goal of the paper is not to present a new benchmark study, but to provide a tool that allows the research community to easily run and reproduce benchmark studies.
>
> ### Writing
>
> We have improved the conciseness and clarity of the writing throughout. A few requested changes are highlighted below:
>
> At L65: "to assess various aspects" is modified to “to assess the fidelity, privacy and utility of the dataset”
>
> At L123: "Synthcity can benchmark … the synthetic data’s utility to downstream tasks." is modified to "Synthcity offers a clean interface so that the user can benchmark the downstream utility of cross-domain generation using only one line of code."
>
> At L138: "Preventing mistakes such as data leakage" is edited to remove the example on data leakage. But to clarify, data leakage may arise in the cross-domain settings because the standard cross-validation functions (e.g. sklearn.model_selection.KFold) are not applicable due to the distribution shift.
>
> Many thanks for raising these points.
>
> ### Relation to prior work
>
> To clarify, the scope of comparison in Table 3 is tabular and time series data generation for downstream ML tasks. Some existing libraries (e.g. SDV) have capabilities beyond this scope. For example,
>
> 1. Ability to handle additional data modalities such as JSON-formatted data and relational databases with multiple tables. These modalities are currently not supported by the Synthcity library.
> 2. Support of specific and highly structured data types such as email addresses, telephone numbers, zip codes, etc. Currently, Synthcity only supports generic data types commonly used in ML tasks (e.g. categorical, numerical, and integer variables).
>
>  Thank you.

---

> ### Author Response · Authors · 2023-08-22
>
> Dear Reviewer,
>
> We are sincerely grateful for your time and energy invested into the review process! Given the limited time left in this response phase, we wanted to check whether our responses have addressed all your concerns. We would be eager to follow up on any additional comments you may have. Thank you!

---

> > ### Comment · Reviewer_T6Ey · 2023-08-22
> > **Thanks for the response**
> >
> > I thank the authors for addressing my concerns adequately. I have raised my score.

---

### Decision · Program_Chairs · 2023-09-22

**Decision:**

Accept (Poster)

**Comment:**

The presented library has a lot of tools to generate synthetic data, and do so with ease. The API appears to be quite generic and can have wide impact. All reviewers are unanimously in favor of the paper, and hence I recommend acceptance.